# Evaluating Prognosis of Gastrointestinal Metastatic Neuroendocrine Tumors: Constructing a Novel Prognostic Nomogram Based on NETPET Score and Metabolic Parameters from PET/CT Imaging

**DOI:** 10.3390/ph17030373

**Published:** 2024-03-14

**Authors:** Yifan Liu, Ruizhe Cui, Zhixiong Wang, Qi Lin, Wei Tang, Bing Zhang, Guanghua Li, Zhao Wang

**Affiliations:** 1Department of Gastrointestinal Surgery, First Affiliated Hospital of Sun Yat-sen University, Zhongshan 2nd Street, No. 58, Guangzhou 510080, China; liuyf269@mail2.sysu.edu.cn (Y.L.); cuiruizhe1997@163.com (R.C.); wangzhx5@mail2.sysu.edu.cn (Z.W.); linq43@mail2.sysu.edu.cn (Q.L.); tangw27@mail2.sysu.edu.cn (W.T.); 2Department of Nuclear Medicine, First Affiliated Hospital of Sun Yat-sen University, Zhongshan 2nd Street, No. 58, Guangzhou 510080, China; zhangb79@mail.sysu.edu.cn

**Keywords:** gastrointestinal, neuroendocrine neoplasm, neuroendocrine tumor, positron emission tomography-computed tomography, prognosis

## Abstract

Introduction: The goal of this study is to compare the prognostic performance of NETPET scores, based on gallium-68 DOTANOC (^68^Ga-DOTANOC) and fluorine-18 fluorodeoxyglucose (^18^F-FDG) Positron Emission Tomography-Computed Tomography (PET-CT), and PET-CT metabolic parameters in metastatic gastrointestinal neuroendocrine tumors (GI-NET), while constructing and validating a nomogram derived from dual-scan PET-CT. Methods: In this retrospective study, G1–G3 GI-NET patients who underwent ^68^Ga-DOTANOC and ^18^F-FDG PET scans were enrolled and divided into training and internal validation cohorts. Three grading systems were constructed based on NETPET scores and standardized uptake value maximum (SUVmax). LASSO regression selected variables for a multivariable Cox model, and nomograms predicting progression-free survival (PFS) and overall survival (OS) were created. The prognostic performance of these systems was assessed using time-dependent receiver-operating characteristic (ROC) curves, concordance index (C-index), and other methods. Nomogram evaluation involved calibration curves, decision curve analysis (DCA), and the aforementioned methods in both cohorts. Results: In this study, 223 patients (130 males; mean age ±  SD: 52.6 ± 12 years) were divided into training (148) and internal validation (75) cohorts. Dual scans were classified based on NETPET scores (D1–D3). Single ^68^Ga-DOTANOC and ^18^F-FDG PET-CT scans were stratified into S1-S3 and F1-F3 based on SUVmax. The NETPET score-based grading system demonstrated the best OS and PFS prediction (C-index, 0.763 vs. 0.727 vs. 0.566). Nomograms for OS and PFS exhibited superior prognostic performance in both cohorts (all AUCs > 0.8). Conclusions: New classification based on NETPET score predicts patient OS/PFS best. PET-CT-based nomograms show accurate OS/PFS forecasts.

## 1. Introduction

Neuroendocrine neoplasms (NENs) are rare heterogeneous tumors from the neuroendocrine cell system, with neuroendocrine differentiation and markers, secreting various peptide hormones and biogenic amines [1]. NENs can occur throughout the body, with the most common being gastroenteropancreatic NENs (GEP-NENs), followed by those in the lungs, and among GEP-NETs, gastrointestinal NETs (GI-NETs) are the most common primary sites [2]. Despite a low incidence rate of 3.56/100,000 per year in the US, there has been a steady rise in recent years [3]. The World Health Organization (WHO) classifies NENs based on hormonal secretion and the presence of hormone-related symptoms into functional and non-functional categories, as well as by their pathological differentiation into well-differentiated neuroendocrine tumors (NETs) and poorly differentiated neuroendocrine carcinomas (NECs) [4].

NENs have a high metastatic potential, with ~20% of patients presenting distant metastases at the initial diagnosis [5], with most patients developing distant metastases during the disease progression [6]. Distant metastases correlate with reduced patient survival, and predicting prognosis for well-differentiated NETs with distant metastasis is challenging due to heterogeneity, presenting a significant clinical challenge.

The WHO grading system, currently the most utilized tool for predicting NEN prognosis, has long been a cornerstone in NEN diagnosis and treatment, significantly guiding clinical decisions [7,8]. However, tumor heterogeneity has complicated the assessment of tumor biology [9], leading to other methods to evaluate the prognoses, such as the use of molecular imaging modalities to depict aggressive cell populations [10].

Positron Emission Tomography (PET) stands as one of the most wildly used molecular imaging method in clinical settings, utilizing tomographic techniques to map the three-dimensional distribution of positron-emitting radiotracers within the body. PET facilitates the noninvasive, quantitative evaluation of biochemical and physiological processes. It is often integrated with Computed Tomography (CT) and Magnetic Resonance Imaging (MRI) technologies to create PET-CT and PET-MRI systems, respectively, allowing for the simultaneous acquisition of molecular metabolic information and anatomical details in a single scanning session [11]. This technology is commonly used for the diagnosis and staging of tumors, as well as for assessing the metabolic activity of tissues [12].

Utilizing various tracers, PET can capture distinct metabolic and biological characteristics of normal and tumor tissues. The most commonly used tracer is the 18F-labelled glucose analogue FDG, whose accumulation in tissues reflects glucose utilization [11]. Tumor activity is linked to the overexpression of GLUT glucose transporters and increased hexokinase activity, making FDG PET widely used in oncology for detection, staging, restaging, and assessing treatment response [13,14].

Another commonly used tracer is the ^68^Ga-DOTA conjugated peptide. The fundamental mechanism of using ^68^Ga-DOTA conjugated peptides to assess tumors expressing somatostatin receptors (SSTRs) is based on these compounds’ high affinity for SSTRs [15,16,17]. SST is a small cyclic neuropeptide present in neurons and endocrine cells, with a high density in the brain, peripheral neurons, endocrine pancreas, and GI tract [11]. Since NENs originate from neuroendocrine cells and most express SSTRs, PET-CT using 68Ga-DOTA conjugated peptides can effectively target and visualize them [18].

PET imaging has emerged as a prominent tool in NEN imaging and guiding optimal systemic therapies selection [19,20]. Research indicates that ^68^Ga-DOTATATE PET/CT exhibits a sensitivity greater than 94% and a specificity greater than 92% for detecting NETs, both superior to conventional CT and MRI [21]. Well-differentiated NEN cells tend to exhibit more SSTRs on their membranes, making PET-CT with ^68^Ga-DOTA an accurate method for identifying well-differentiated NENs [22]. Positive results from these imaging techniques usually indicate lower tumor aggressiveness, a better prognosis, and that patients are more suitable candidates for Peptide Receptor Radionuclide Therapy (PRRT) [23,24]; meanwhile, increased avidity on ^18^F-FDG PET indicates higher metabolic activity, suggesting aggressive biology and poorer prognosis. At this point, PRRT becomes less applicable [25,26,27].

Many studies confirm a significant link between ^18^F-FDG and ^68^Ga-DOTA PET/CT scan metabolic parameters (e.g., SUVmax) and GEP-NET prognosis [25,28]. Recent research shows that combining both tracers in PET/CT scans improves prognostic insights for metastatic GEP-NET patients. The most notable is Chan, David L. et al.’s NETPET scoring system [29], which has been found to independently correlate with prognosis and is widely recognized [30,31].

Previous studies have primarily investigated the association between individual assessment methods and patient prognosis, with few comparing the predictive performance of these methods. Additionally, they only demonstrated the relationship between PET-CT-related parameters or scores and NET prognosis without extensively utilizing them for prognostic prediction.

Therefore, our present study aims to compare the prognostic prediction efficacy of NET-PET scores and PET-CT metabolic parameters and to establish a novel prognostic tool by combining them with clinical pathological indicators.

## 2. Results

### 2.1. Patient Characteristics

We selected patients based on the inclusion and exclusion criteria as previously mentioned (shown in Appendix A). Ultimately, 223 consecutive patients (Male, 130 patients) with distant metastases from GI-NETs were included. Primary tumors originated in the stomach (33, 14.8%), small intestine (51, 22.9%), and colon and rectum (139, 62.3%). Of these, 105 (47.1%) did not undergo surgery, while 118 (52.9%) had primary tumor resection. All primary lesions were pathologically confirmed: G1 (42, 18.8%), G2 (161, 72.2%), and G3 (20, 9%). The average FDG SUVmax value was 4.32 (SD: 4.09) and the median was 13.5 (range: 0–49.3). The average SSA SUVmax value was 14.0 (SD: 9.42) and the median was 26 (range: 2–63.0). The majority of patients (86 patients, 38.6%) were classified as P2b based on the NETPET score. Further baseline characteristics and NETPET scores can be found in Table 1.

### 2.2. Construction of the Novel Grading Systems

Using X-tile software, ^18^FDG-SUVmax, ^68^Ga-DOTANOC-SUVmax, and NETPET scores were all divided into three categories (shown in Appendix A). Based on NETPET scores, 92 patients (41.3%) were classified as D1 (SSTRI+ve/FDG-ve and SSTR uptake > FDG uptake, 1–2 lesions, equal to NETPET score P1–P2a), 86 (38.6%) as D2 (SSTRI uptake > FDG uptake, >2 lesions, equal to NETPET score P2b) and 45 patients (20.2%) as D3 (SSTRI uptake <= FDG uptake and SSTRI-ve/FDG+ve, equal to NETPET score P3-P5). For 18F-FDG PET-CT semiquantitative grades, 115 patients (51.6%) were F1(FDG SUVmax < 3.5), 62 (27.8%) F2 (FDG SUVmax: 3.5–6.1), and 46 (20.6%) F3 (FDG SUVmax > 6.1). For ^68^Ga-DOTANOC PET-CT semiquantitative grades, 21 patients (9.4%) were S1(SSA SUVmax < 5), 55 (24.7%) S2 (SSA SUVmax: 5–8.1), and 147 (65.9%) S3 (SSA SUVmax > 8.1). The flowchart for the D grading process and representative imaging examples for the three grading systems can be seen in Figure 1.

### 2.3. Overall Survival and Progression-Free Survival for Patients

During follow-up, 68 (30.5%) patients died, and 195 (87.5%) patients were found to have disease progression or death. The mean follow-up time was 27.2 ± 12.4 months (median follow-up time: 26 months, range: 2–63 months). Survival analysis was performed on PFS and OS using four grading systems with the Kaplan–Meier (KM) method. The results showed that the D grading system had the most excellent discriminative ability for both OS and PFS. The median OS by the D grading system was unreached for D1, 48 months for D2, and 22 months for D3 (*p* < 0.001, shown in Figure 2a). The median PFS by D grading system was 23 months for D1, 12 months for D2, and 6 months for D3 (*p* < 0.001, shown in Figure 2e). Similarly, the F grading system also exhibited remarkable discriminatory ability for both OS and PFS. The median OS by F grading system was 60 for F1, 35 months for F2, and 24 months for F3 (*p* < 0.001, shown in Figure 2b). The median PFS by F grading system was 18 months for F1, 11 months for F2, and 6 months for F3 (*p* < 0.001, shown in Figure 2f). The S classification system exhibits poor discriminative ability for both OS (*p* = 0.16, shown in Figure 2c) and PFS (*p* = 0.43, shown in Figure 2g). The WHO G grading system, however, demonstrated only moderate discriminatory ability for both OS (*p* = 0.52, Figure 2d) and PFS (*p* = 0.05, shown in Figure 2h).

### 2.4. Comparing the Prognostic Value of D Grade, F Grade, S Grade, and WHO Grading System

Index calculations for OS and PFS prediction were conducted across three models and the WHO grading system, revealing the highest values for D grade (D vs. F vs. S vs. WHO; OS: 0.763, 0.727, 0.566, 0.650; PFS: 0.724, 0.630, 0.556, 0.592). Additionally, D grade achieved the lowest AIC, highest LR-test, and R2 values for both OS and PFS, indicating superior model fit (shown in Table 2). ROC curves (shown in Appendix A), AUC, NRI, and IDI analyses further demonstrated enhanced overall predictive performance and clinical utility of D grade for OS and PFS (with D grade as a reference, NRI and IDI for other gradings were <0, except for FDG in the 1-year OS, whose NRI was >0, but the *p*-value was >0.1, shown in Appendix A).

### 2.5. Construction and Validation of the Nomograms

The entire cohort was randomly divided into a training set and internal validation set at a 2:1 ratio. Ultimately, 148 patients were incorporated into the modeling set for constructing prognostic models for OS and PFS, while 75 patients were included in the internal validation set to assess the predictive performance of these prognostic models. There were no statistically significant differences in baseline characteristics and D, F, and S classifications between the two groups (all *p* > 0.05).

Using LASSO regression and cross-validation in the training cohort (shown in Appendix A), five OS-predicting variables were identified: age, metastasis status, WHO G grade, D grade, and F grade. These variables informed prognostic nomograms for patients’ 1-, 2-, and 3-year OS (shown in Figure 3a). Multivariate Cox regression analyses revealed D classification as independently and significantly associated with better OS both in the training cohort and internal validation cohort (shown in Table 3). Similarly, LASSO regression identified six PFS-predicting variables (shown in Appendix A): age, metastasis status, treatment modality, WHO G grade, D classification, and F classification. Nomograms for patients’ 6-, 12-, and 18-month PFS were constructed (shown in Figure 3b). Multivariate Cox regression analyses revealed D classification as independently and significantly associated with better PFS both in the training cohort and internal validation cohort (shown in Table 4).

### 2.6. Validating and Comparing the Prognostic Value of Nomogram, D Grading System, F Grading System, and WHO Grading System

The nomograms were validated for their performance in both the training set and the internal validation set. Initially, the nomograms exhibited a C-index of 0.810 (95% CI: 0.747–0.874) for OS prediction within the training set and a C-index of 0.741 (95% CI: 0.692–0.789) for PFS prediction. In the internal validation set, the C-index for OS prediction was 0.849 (95% CI: 0.781–0.916), while the C-index for PFS prediction was 0.824 (95% CI: 0.778–0.871). The ROC curves for 1-, 2-, and 3-year OS predictions demonstrated an AUC of over 0.8 in both the training and internal validation sets (shown in Figure 4a–c,g–i). Similarly, the ROC curves for 6-, 12-, and 18-month PFS predictions showed an AUC of over 0.8 in both sets (shown in Figure 4d–f,j–l). Additionally, the calibration curves indicated a good fit between the nomogram predictions and the actual event occurrences (shown in Figure 5).

Furthermore, the DCA demonstrated upward trends for both OS and PFS nomograms, with excellent separation from the baseline models. The DCA curves exhibited higher net benefits within the common threshold selection range, regardless of whether they were applied to the training cohort or the internal validation cohort (shown in Figure 6).

We evaluated the nomogram’s prognostic efficacy against D, F, and WHO classifications. The ROC curves showed nomograms had superior 1-, 2-, and 3-year OS, and 6-, 12-, and 18-month PFS predictions in both training and validation cohorts, with higher AUC values (shown in Figure 4). DCA revealed the nomograms had the highest net benefit across most threshold probabilities (shown in Figure 6). The goodness of fit was assessed, and the nomograms showed the lowest AIC and highest C-index, R-squared, and LR test values for OS and PFS predictions in both cohorts (shown in Table 5 and Table 6). Clinical applicability was assessed using NRI and IDI metrics, with the nomogram outperforming other staging systems in both cohorts (shown in Appendix A).

In our final analysis, we divided the training and internal validation sets into several subgroups based on different primary tumor locations and treatment modalities. This allowed us to validate the nomogram’s efficacy across these diverse subgroups. The ROC curves, as shown in Appendix A, indicate that the model consistently achieved predictive accuracies for both OS and PFS with AUCs of ≥0.75 across all subgroups of different primary locations (illustrated in Appendix A). Further validation of the model’s performance through the calculation of the C-index yielded an average C-index of 0.84 (as detailed in Appendix A). In the subgroups categorized by different treatment modalities, the ROC curves (illustrated in Appendix A) demonstrate that the model’s predictions for both OS and PFS remained robust with AUCs of ≥0.79, and the average C-index was calculated to be 0.80 (as detailed in Appendix A). This evidence supports the model’s stability and applicability across varied clinical subgroups.

### 2.7. Establishment of New Risk Classification and Online Models for Convenient Clinical Use

We calculated nomogram-derived scores for OS and PFS, stratifying them using X-tile software (Appendix A). OS risk stratification: high (>190 points), medium (136–190 points), low (<136 points); PFS: high (>170 points), medium (115–170 points), low (<115 points). Nomogram-based risk stratification outperformed D, F, and WHO staging in both cohorts (Appendix A). Furthermore, two online interactive nomograms were developed for clinicians: OS prediction (https://fah-sysu-gastrointestinal-surgery.shinyapps.io/NOMOOS/ or Figure 3a QR code) and PFS prediction (https://fah-sysu-gastrointestinal-surgery.shinyapps.io/NomoPFS/ (accessed on 8 March 2024) or Figure 3b QR code).

## 3. Discussion

Over the past two decades, medical imaging technology has experienced significant advancements, including remarkable progress in nuclear medicine. This is particularly evident in the context of NETs. As the majority of well-differentiated NETs express somatostatin receptors, the use of PET-CT imaging with specialized radiotracers demonstrates unique advantages [32]. Numerous studies have reported that PET-CT imaging using ^68^Ga-labeled somatostatin analogs (^68^Ga-SSA) as contrast agents exhibit exceptionally high specificity and sensitivity in the diagnosis of NETs [33,34,35]. Additionally, ^68^Ga PET/CT can serve as a valuable tool in guiding the management of Peptide Receptor Radionuclide Therapy (PRRT) and Somatostatin Analogues (SSA) [36]. Therefore, it plays a critical role in the comprehensive management of NETs.

^18^F-FDG is the most commonly used radiotracer in clinical PET-CT imaging. It is effective in detecting a wide range of tumors with high sensitivity to glucose metabolism and can reveal their metabolic activity, thereby indicating the malignancy level of the tumor [37,38]. In the context of NETs, ^18^F-FDG PET-CT imaging plays a crucial role in the diagnosis and detection of tumors with higher proliferative potential.

Not all NETs are suitable for PET/CT scans, as excessive use of nuclear medicine examinations may lead to radiation damage in patients and resource wastage. According to the consensus of the European Association of Nuclear Medicine (EANM) [39], ^68^Ga-DOTA-based PET/CT is only applicable for 1. NETs with an unknown primary site, 2. metastatic NETs, and 3. staging/restaging of NETs. Meanwhile, ^18^F-FDG-based PET/CT is limited to: 1. neuroendocrine carcinoma; 2. G3 NETs; and 3. G1 and G2 NETs with confirmed negative somatostatin PET/CT uptake.

Although there are differences in the roles and scope of application for ^68^Ga-DOTANOC PET/CT and ^18^F-FDG PET/CT in the diagnosis and treatment of NETs, numerous studies have demonstrated that the relationship between the two modalities should be viewed as complementary rather than competitive. Irfan Kayani et al. [40] collected imaging results from 38 patients diagnosed with neuroendocrine tumors (NETs) and analyzed the diagnostic efficacy of using ^68^Ga-DOTATATE and ^18^F-FDG PET/CT scans individually as well as in combination. The findings indicated that the sensitivity and specificity of the combined dual-tracer scans were higher than those of either tracer alone in diagnosing NETs (dual-tracer sensitivity: 92%, ^68^Ga-DOTATATE PET/CT sensitivity: 82%, and ^18^F-FDG PET/CT sensitivity: 66%). Duygu Has Simsek et al. [31] investigated the relationship between the maximum SUVmax of ^68^Ga-DOTATATE and ^18^F-FDG PET/CT and their correlation with histopathological findings and metastasis. The results revealed that the combined impact of ^18^F-FDG and ^68^Ga-DOTATATE PET/CT on treatment decision making was 59%. Furthermore, the dual-tracer scans could overcome the limitations of histopathological grading, particularly in intermediate-grade GEP-NETs.

In recent years, the prognostic implications of dual scans have increasingly gained attention. Among these, the NETPET scoring system, pioneered by Chen et al. [29], is the most widely accepted system at present. It primarily targets metastatic GEP-NET and stratifies patients’ prognostic risk based on the comparison of SSTR and FDG uptake in lesions on whole-body PET-CT scans, along with the number of metastatic lesions. The system is divided into eight categories, from P1 to P5. Its prognostic stratification effectiveness has been validated in multiple centers [41] and has proven to be effective beyond GEP-NETs [42].

Our study’s dual-scan visual classification is based on the detailed NETPET scoring system. Previous research typically divides NETPET scores into three categories: P1, P2–P4, and P5, without exploring subdivisions within P2–P4. We found prognosis worsened with increasing lesions when SSTRI uptake surpassed FDG uptake in the ‘referred lesion’. We incorporated lesion count from PET-CT into our study, classifying as D1 (SSTRI+ve/FDG-ve and SSTR uptake> FDG uptake, 1–2 lesions), D2 (SSTRI uptake > FDG uptake, >2 lesions), and D3 (SSTRI uptake ≤ FDG uptake and SSTRI-ve/FDG+ve). This stratification is justified, as our cohort mostly consists of well-differentiated patients (G1-G2, 91%), who generally have better prognoses. In these patients, the tumor burden plays a crucial role in prognosis.

Recently, numerous studies have explored the prognostic implications of semi-quantitative or quantitative data derived from PET/CT scans. Amit Tirosh et al. [43] assessed the relationship between total ^68^Ga-DOTATATE-avid tumor volume (^68^Ga-DOTATATE TV) and progression-free survival (PFS), finding that the quartiles of ^68^Ga-DOTATATE TV were negatively correlated with PFS (*p* = 0.001) and disease-specific survival rates (*p* = 0.002). This demonstrated that the ^68^Ga-DOTATATE TV values were associated with the prognosis of NET patients. Additionally, Akira Toriihara et al. [44] investigated the association between the maximum SUVmax in lesions with the highest ^68^Ga-DOTATATE uptake for each patient and PFS, revealing a significant correlation between the two. J. Zhang et al. [45] included 495 patients with metastatic GEPNETs who underwent PRRT. They investigated the relationship between ^18^F-FDG PET/CT SUVmax values and qualitative indicators with OS and PFS. The results revealed that the presence of ^18^F-FDG PET-positive lesions was an independent prognostic factor for NEN patients receiving PRRT treatment. It is worth noting that whole-body PET-CT imaging offers unique advantages over other imaging modalities in detecting various tumor metastatic lesions throughout the body [46].

In this study, like most existing research, we utilized SUVmax as a semi-quantitative indicator for single PET/CT scans. A new grading system was developed based on SUVmax, incorporating an F grading system for ^18^F-FDG and an S grading system for ^68^Ga-DOTA We compared F grading, S grading, and a combined dual-scan D grading using the KM method. The results demonstrated that both the combined ^68^Ga-DOTA and ^18^F-FDG PET/CT scans (median survival: D1 vs. D2 vs. D3, 60 months vs. 48 months vs. 22 months, as shown in Figure 1a; median PFS: D1 vs. D2 vs. D3, 23 months vs. 12 months vs. 6 months, as shown in Figure 1e) and the single ^18^F-FDG PET/CT scans (median survival: D1 vs. D2 vs. D3, 60 months vs. 35 months vs. 24 months, as shown in Figure 1b; median PFS: D1 vs. D2 vs. D3, 18 months vs. 11 months vs. 9 months, as shown in Figure 1f) provided good stratification for patients’ OS and PFS. Furthermore, the KM curves indicated that higher D and F gradings are associated with worse prognosis, which is logical since higher D gradings indicate an increase in lesion count and glucose uptake capacity over somatostatin. High F grading signifies strong glucose uptake by lesions, a marker of high tumor malignancy. The KM analysis enlightens us that in evaluating PET/CT scans of NENs patients, a propensity for lesions to uptake more glucose than somatostatin is an indicator of poor prognosis, warranting attention from clinicians.

These results are similar to the studies by Tina Binderup [25]. They used SUVmax values as semi-quantitative indicators for ^18^F-FDG PET/CT to explore their relationship with prognosis and discovered that FDG SUVmax was significantly correlated with patient prognosis, consistent with our study results. In their research, the cutoff point for FDG-positive SUVmax was set at 4, while the maximum survival difference was found at a cutoff of 7 (*p* = 0.001), similar to our study’s stratification for FDG SUVmax values (FDG SUVmax D1, 0–3.5, D2: 3.6–6.1, D3 ≥ 6.1).

We further compared the stratification and prediction performance of the three classifications for patient prognosis in this research. The results revealed that the D classification provided the best stratification and prediction performance for both OS and PFS, followed by the F classification, both of which outperformed the WHO classification; the S classification performed the worst. Our findings are in line with those of Tina Binderup et al. [25], who observed that FDG positivity/negativity offered better prognostic risk stratification than histopathological grading. Hwan Lee et al. [47]. also found that dual-tracer stratification based on SUVmax values could reflect G3 tumor characteristics and serve as an alternative to histopathological grading. This implies that if feasible, using dual-scan grading or single ^18^F-FDG-PET/CT semi-quantitative parameters can serve as non-invasive prognostic factors, and their performance is superior to histopathological grading, which requires an invasive approach.

In addition to pathological grading, there are other commonly used clinical prognostic markers for NETs, such as chromogranin A (CgA) and the liquid biopsy-based NETest. Similar to the NETPET scoring system, they can also indicate the PFS or OS for patients [48,49,50,51]. In a study covering 152 patients with GEP-NETs, the NETest reached a 12-month PFS AUC value of 0.78, while CgA had an AUC of 0.73 [52]. In another study evaluating whether NETest could identify the potential of disease recurrence after NET resection, the accuracy of the NETest prediction marker was 94%, the specificity was 92%, and the sensitivity was 100% until the exact time of recurrence or 24 months, whereas the accuracy of the CgA prediction marker was 57%, with a specificity of 76% and a sensitivity of 15%, failing to reach the accurate diagnostic threshold [53]. In our study, the D grading system achieved a 12-month PFS AUC value of 0.81 in independent internal validation. Considering its non-invasive nature, it can even be comparable to NETest and better to CgA, which indicates strong prognostic performance and clinical application potential. Additionally, the D grading system possesses strong interpretability, unlike CgA, which, despite demonstrating diagnostic and prognostic value across various diseases in clinical settings, has not always produced satisfactory results [54]. It has the potential to replace CgA in the future.

Within this research, a novel nomogram incorporating D and F gradings along with clinicopathological factors was developed to predict OS and PFS in patients with GI-NENLM. The nomogram’s predictive performance and consistency were validated using an independent internal validation cohort, outperforming D and F gradings as well as histological classifications in terms of prediction accuracy and clinical utility. This fills a gap in the prediction of patient prognosis in the metastatic GEP-NET domain. Previous studies often utilized public data to predict surgery-related outcomes [55,56]. Our research addresses the unmet need for prognostic models specific to metastatic GEP-NETs.

Previous studies have also designed prognostic models for NETs. In a study based on the SEER database targeting patients with GEP-NENs, the nomogram they constructed for OS yielded C-indexes of 0.821 and 0.823 in the primary and validation groups, respectively [57]. Additionally, in a study focusing on patients with GEP-NENs and liver-limited metastasis, the nomogram for OS achieved C-indexes of 0.814, 0.826, and 0.789 in the training, internal validation, and external validation sets, respectively [58].These results are closely aligned with ours, which are 0.810 and 0.849, yet they lacked an investigation into patients’ PFS, which is crucial for a disease characterized by a relatively mild and chronic course, such as NETs. In our study, we also constructed a predictive model for patients’ PFS and obtained favorable results. Furthermore, this is the first study to incorporate PET-CT visual scoring and related parameters into a clinical pathological factor model. Future research should further include circulating biomarkers to enhance the model’s predictive capability. While this model demonstrates excellent predictive performance, it is important to consider the strong heterogeneity inherent in NETs when applying it in clinical settings. Consequently, the predictive model should be viewed as an adjunct tool, complementing clinical guidelines and the MDT approach. This integration enables more precise and flexible clinical applications, tailored to the unique context of each patient’s case.

Additionally, we’ve established an online prediction platform (Figure 2) and a grading system based on nomogram scores to streamline clinicians’ assessments and prognostic evaluations. In clinical usage, for example, a patient meeting the inclusion criteria may receive D, F, and S grades with the assistance of a nuclear medicine physician during treatment. By integrating these grades with pathological grading and other clinical circumstances, our predictive model can be utilized to assess their prognosis. If a patient’s pathological grade is G3, with D and F grades exceeding D2 and F2, respectively (indicating high FDG uptake by the tumor), it is likely the patient may have a poor prognosis with a 2-year survival rate of less than 50% and a high risk of disease progression in the near term. This would necessitate more aggressive clinical treatment and follow-up. Conversely, if the patient has a better tumor differentiation (G1–G2) and nuclear medicine assessment grades are D1 and F1 (indicating stronger uptake of somatostatin by the tumor), then the patient’s prognosis is likely to be better, with a 3-year survival rate reaching up to 90%, and the tumor progression being slow, which may call for a more conservative approach to treatment and monitoring.

Our study has some limitations: it used single-center data from a small patient sample for developing grading models and prognostic nomograms, requiring external validation. The high cost of PETCT and radiation exposure limit the number of patients. We assessed lesion sites using dual-scan PET-CT imaging with NETPET scoring criteria, the D grading system based on the NETPET score may be influenced by the quality of imaging technology, the experience of operators, the difference in image protocols and the consistency of imaging interpretation standards. Therefore, its practical application in assisting clinical decision making requires further research and validation, integrating it with patients’ specific clinical situations for personalized interpretation and application, involving visual assessment methods that may require expert interpretation. For single-scan evaluation, we only used SUVmax, excluding other PET-CT parameters like Metabolic Volumetric Index (MVI) and Mean Parenchymal Volume (MPV). The retrospective nature of this study limits our control over baseline patient characteristics and only including GI-NET may limit the generality of the model. Despite these limitations, our conclusions remain significant. Future research should involve multi-center, large-scale, prospective studies combining deep learning and PET-CT radiomics for better prognostic guidance in GI-NET patients with metastases. Additionally, we should also initiate prospective studies comparing the prognostic accuracy of the NETPET score with circulating biomarkers and incorporate these markers into our prognostic model to enhance its potential and functionality.

## 4. Materials and Methods

We analyzed all patients pathologically diagnosed with well-differentiated (G1–G3) GI-NETs at The First Affiliated Hospital of Sun Yat-sen University between 2016 August and 2021 August. Eligible participants are: (1) aged 18–75 years; (2) with pathologically confirmed GI-NENs; (3) evidence of distant metastasis supported by radiologic or pathologic data; (4) undergone concurrent PET-CT scans using ^18^F-FDG and ^68^Ga-DOTA radiotracers, excluding individuals: (1) classified with NEC per WHO pathological grading; (2) with treatment of PRRT within 1 year; (3) with intervals exceeding 30 days between PET-CT scans; (4) experiencing death or disease progression within 3 months; (5) with incomplete diagnostic, therapeutic, or radiologic information; (6) NENs originating from non-GI locations or with unknown primary sites. If a patient has undergone multiple rounds of PET-CT scans, select the initial pair for analysis.

### 4.1. PET/CT Imaging Information Acquisition and Analyses

All patients signed informed consent forms for PET-CT examinations. Before ^68^Ga-DOTANOC PET-CT and ^18^F-FDG examination, fasting for 6 h was required for the patients. PET-CT imaging was performed with a Gemini GXL 16 PET scanner (Philips Healthcare, Andover, MA, USA). Between 111 and 185 MBq (3–5 mCi) ^68^Ga-DOTANOC or a dose of 5.18 MBq (0.14 mCi)/kg FDG was injected intravenously and scanned continuously. Approximately 45–60 min after the injection, serial scanning was performed from head to midthigh. Following low radiation dose CT acquisition with a slice thickness of 5 mm, the PET acquisition was performed for 1.5 min per bed position for 7–8 beds using a slice thickness of 4 mm. CT-based attenuation correction of the emission data was employed. PET images were reconstructed by the Line of Response RAMLA algorithm. The interval between ^18^F-FDG and ^68^Ga-DOTANOC PET-CT studies shall be at least 24 h.

After undergoing training on the visual evaluation and scoring of ^68^Ga-DOTANOC and ^18^F-FDG PET-CT images according to the NETPET scoring system [29,41], two nuclear medicine experts independently scored the patients using a blinded method. In cases of discrepancy, Professor Zhao Wang, an expert with 30 years of experience in diagnosing and treating NETs, made the final decision. To be detailed, the NETPET score is classified into grades 0–5, which is in total 9 levels. Due to limitations in quantity, P0 is typically not included in the analysis. When a patient’s lesions exhibit uptake of SSTRI without FDG uptake, they are classified into grade P1. Conversely, lesions with FDG uptake but without SSTRI uptake are classified as grade P5. Grades P2–P4 represent lesions that have concurrent uptake of both SSTRI and FDG, without isolated FDG uptake. The more detailed classification requires counting the number of lesions that meet these criteria to define the specific grade. When selecting and comparing lesions, all lesions that have been identified as tumors should be included. Identify the single lesion with the most FDG-avidity relative to its SSTR uptake, which should be selected as a reference, and its size relationship with self-SSTR uptake should be compared. When a patient has multiple records of PET-CT scans, the earliest one is selected. In short, this system not only considers the comparison of FGD and SSTRI uptake of tumor lesions, but also considers the number of lesions, which is the most comprehensive scoring system for dual scanning so far. The PET/CT images were evaluated visually qualitatively and semi-quantitatively. When the activity in each area exceeded background levels and could not be attributed to physiological activity, it was identified as tumor tissue. The evaluators were blinded to the findings of the structural imaging. Any non-physiological focus of ^68^Ga-DOTANOC uptake above background was considered abnormal. Likewise, on ^18^F-FDG PET/CT images, any non-physiological focus of ^18^F-FDG uptake greater than the background blood-pool activity or adjacent normal tissue was considered positive. The PET/CT images were combined with non-enhanced CT to obtain anatomical 3D imaging. Metastatic disease was categorized into liver metastases and extrahepatic metastases.

PET-CT imaging offers a variety of semi-quantitative metrics, including Standardized Uptake Value (SUV), Metabolic Volumetric Index (MVI), and Mean Parenchymal Volume (MPV) and so on. Extensive research has indicated that the SUV of the primary tumor has been investigated as a potential prognostic factor for survival [59,60]. Among these, the most frequently utilized metric is SUVmax, which represents the maximum SUV within the tumor and is considered a key indicator of tumor metabolic activity. This high uptake value is often associated with increased tumor aggressiveness and poor prognosis, making SUVmax a valuable tool in evaluating cancer severity and potential outcomes [25,61]. Therefore, in this study, the SUVmax of both primary and metastatic lesions was calculated one hour post-injection of the contrast agent when evaluating single ^68^Ga-DOTANOC and ^18^F-FDG PET-CT images. These measurements were then utilized as the respective semi-quantitative metabolic parameters for each scanning modality in the subsequent development of a prognostic model. All the SUVmaxs were standardly calculated using the default method in relation to body weight and injected doses: (target tissue(gram)/injected dose(megabecquerels)/body weight(gram).

In our study, we employed the open-source X-tile software (https://medicine.yale.edu/lab/rimm/research/software/ (accessed on 8 March 2024)) to identify optimal cutoff points for NETPET scores, SUVmax values in ^18^F-FDG PET-CT, and ^68^Ga-DOTANOC PET-CT imaging, leveraging its widespread use in determining variable thresholds [62,63,64,65]. These parameters were stratified into three tiers based on the maximized Chi-square values for overall survival, as determined by the Kaplan–Meier method. These newly established classifications were named as follows: 1. Dual-scan grading (D grade); 2. ^18^F-FDG PET-CT semi-quantitative grading (F grade); 3. ^68^Ga-DOTANOC PET-CT semi-quantitative grading (S grade).

### 4.2. Treatment and Follow-Up

All patients’ treatment plans were decided after discussion by our hospital’s NETMDT team. Treatment modalities were categorized into surgical treatment (primary tumor resection performed) and medical treatment (primary tumor resection not performed). There are two primary endpoints in this study: overall survival (OS) and progression-free survival (PFS). OS is defined as between the initial diagnoses of the disease and death from any cause or the date of the last follow-up for patients who are still alive. PFS is defined as the length of time during which a patient’s disease does not progress or worsen after treatment, recorded according to the Response Evaluation Criteria in Solid Tumors version 1.1 (RECIST 1.1) [66].

### 4.3. Statistical Analyses

We evaluated the D, S, and F grading systems’ predictive performance and clinical utility for patients’ PFS and OS in the overall cohort, comparing them to histological grading. We employed time-dependent ROC curves to assess accuracy and specificity, Harrell’s C-index for discrimination power, and NRI and IDI with Z tests for comparing grading systems. For OS prediction, NRI and IDI calculations employed risk categories of (0–0.1, 0.1–0.4, 0.4–0.6, 0.6–1). Likewise, for PFS prediction, risk levels were categorized as (0–0.2, 0.2–0.5, 0.5–0.7, and 0.7–1) in NRI and IDI calculations. Lastly, we used AIC, R-squared, and LR tests to determine the goodness of fit for various models predicting PFS and OS.

In a 2:1 ratio, the overall cohort was randomly divided into a training set and internal validation set. Baseline characteristics and clinical pathological features were compared between the two groups using *t*-tests for continuous variables, chi-squared tests for categorical variables, Fisher’s exact test, and the Mann–Whitney U test. Continuous variables were reported as either the mean with standard deviation (SD) or the median with interquartile range (IQR).

LASSO regression introduces L1 regularization to decrease model complexity, effectively preventing overfitting and facilitating variable selection. It has been widely adopted in building predictive models within the medical field [67,68]. In this study, LASSO-Cox regression was utilized within the modeling group, targeting OS and PFS endpoints, to screen for all potential variables that could be prognostic factors. Specifically, the method of cross-validation was employed, selecting variables based on the lambda minimum criterion. This approach helps in identifying the most relevant predictors by applying a penalization to the model’s coefficients, effectively reducing the risk of overfitting and enhancing the model’s predictive accuracy for patient outcomes.

Univariate and multivariate Cox regression were used to examine associations with survival. Subsequently, nomograms predicting 1-, 2-, and 3-year OS and 6-, 12-, and 18-month PFS were established based on the multivariate survival risk model factors. To assess the nomograms’ predictive ability, we applied the aforementioned evaluation methods, calibration curves, and decision curve analysis (DCA) in the training and internal validation sets independently. The predictive performance of the nomograms with that of the WHO histological grading system was also compared.

To further validate the model’s generalizability and practicality, we performed a subgroup analysis based on the primary site and different treatment modalities. We utilized time-dependent ROC curves and the C-index to conduct sensitivity analysis of the model across different subgroups.

### 4.4. The Construction of a Portable Nomogram

To enhance clinical utility, we employed two strategies: (1) Developed an online prognostic app accessible via a URL or QR code, allowing users to input parameters for 1-, 2-, 3-year OS or 6-, 12-, 18-month PFS predictions; and (2) calculated nomogram scores for each patient and stratified them into three risk groups (high, medium, low) for OS and PFS using X-tile (3.6.1) software, enabling efficient prognosis assessment.

Statistical analyses were conducted with R software (4.2.0), SPSS 26, Xtile (3.6.1), Microsoft Excel (2023), and Adobe Acrobat (2022). A *p*-value < 0.05 was considered significant. Tables and figures were created using the aforementioned software.

## 5. Conclusions

In summary, our study found that the visual grading based on both ^68^Ga-DOTA-NOC and ^18^F-FDG PET-CT scans can better predict patient prognosis compared to the WHO histological grading and single scans. This approach has the potential to improve the prognostic risk stratification of well-differentiated (G1–G3) metastatic gastrointestinal neuroendocrine tumors within the WHO grading system. We developed prognostic nomograms based on PET-CT visual and semi-quantitative data grading, which demonstrate good predictive performance for both OS and PFS. Additionally, we designed and created an online dynamic nomogram and a new risk stratification system, enabling clinicians to quickly assess the prognosis of their target patients and guide subsequent treatments.

## Figures and Tables

**Figure 1 pharmaceuticals-17-00373-f001:**
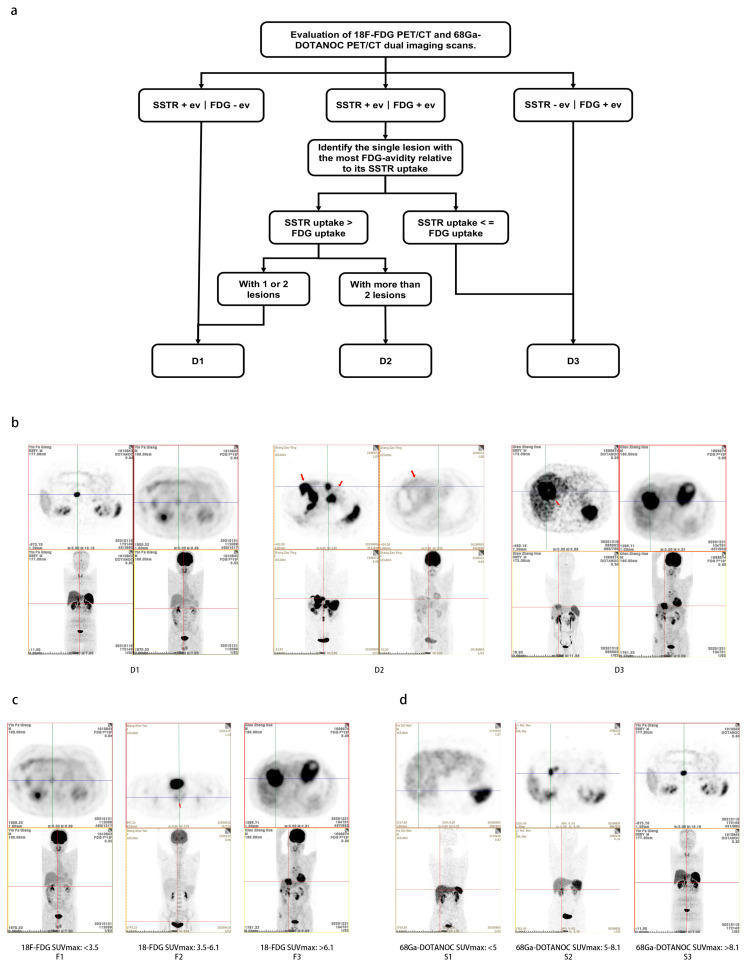
The flowchart of dual ^18^F−FDG and ^68^Ga−DOTANOC PETCT visual grade (D grade) and examples of D grade, semiquantitative ^18^F-FDG PETCT grade (F grade) and semiquantitative ^68^Ga-DOTANOC PETCT grade (S grade). (**a**) The flowchart of D grading system. (**b**) Examples of the D grading system. (**c**) Examples of the F grading system. (**d**) Examples of the S grading system.

**Figure 2 pharmaceuticals-17-00373-f002:**
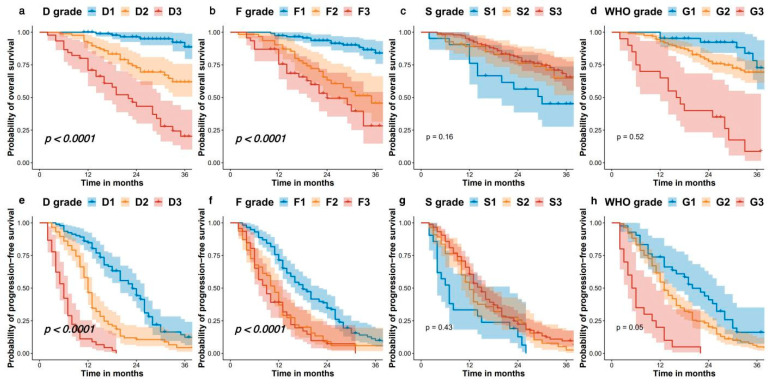
The Kaplan–Meier curves for OS and PFS for the dual ^18^F-FDG and ^68^Ga DOTANOC PETCT visual grade (D grade), semiquantitative ^18^F-FDG PETCT grade (F grade) and semiquantitative ^68^Ga-DOTANOC PETCT grade (S grade). (**a**) Kaplan–Meier curves for OS stratified by D grade. (**b**) Kaplan–Meier curves for OS stratified by F grade; (**c**) Kaplan–Meier curves for OS stratified by S grade. (**d**) Kaplan–Meier curves for OS stratified by WHO grade; (**e**) Kaplan–Meier curves for PFS stratified by D grade. (**f**) Kaplan–Meier curves for PFS stratified by F grade. (**g**) Kaplan–Meier curves for PFS stratified by S grade. (**h**) Kaplan–Meier curves for PFS stratified by WHO grade.

**Figure 3 pharmaceuticals-17-00373-f003:**
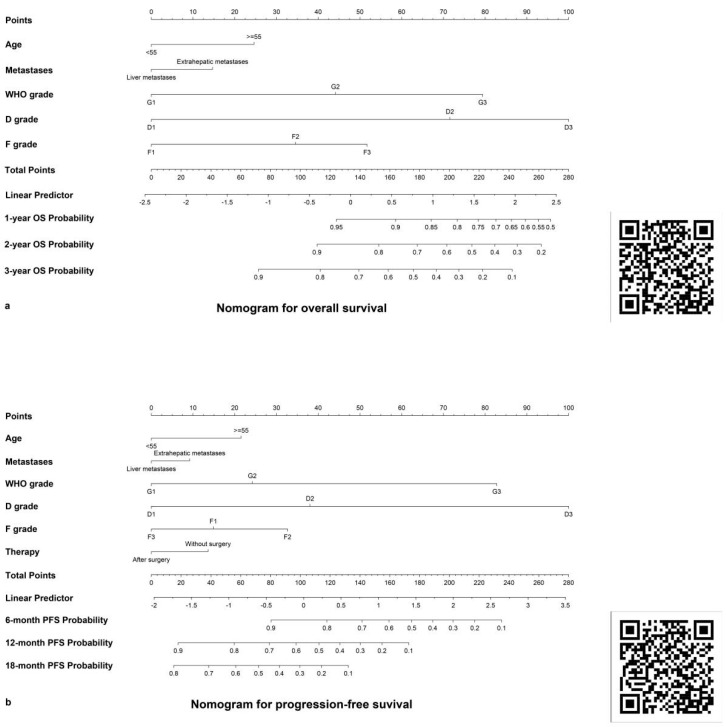
Nomograms for predicting OS and PFS. (**a**) Nomogram for predicting 1-, 2-, 3-year OS. (**b**) Nomogram for predicting 6-, 12-, 18-month PFS.

**Figure 4 pharmaceuticals-17-00373-f004:**
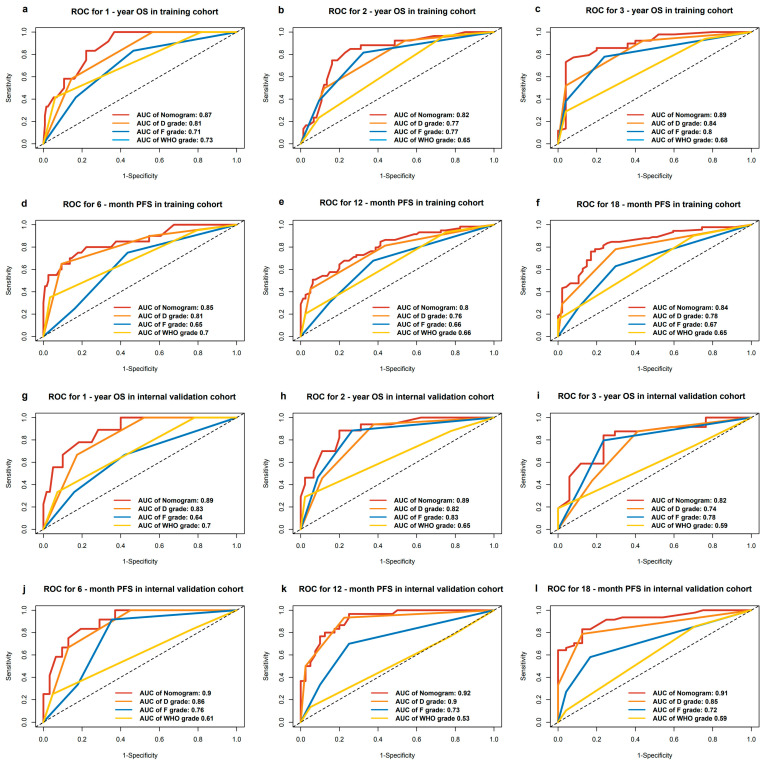
ROC curves of the nomograms, dual 18F-FDG and 68Ga DOTANOC PETCT visual grade (D grade), semiquantitative 18F-FDG PETCT grade (F grade) and WHO pathological grading system (G grade) (**a**–**c**) ROC curves for predicting 1-, 2-, 3-year OS in training cohort. (**d**–**f**) ROC curves for predicting 6-, 12-, 18-month PFS in training cohort. (**g**–**i**) ROC curves for predicting 1-, 2-, 3-year OS in internal validation cohort. (**j**–**l**) ROC curves for predicting 6-, 12-, 18-month PFS in internal validation cohort.

**Figure 5 pharmaceuticals-17-00373-f005:**
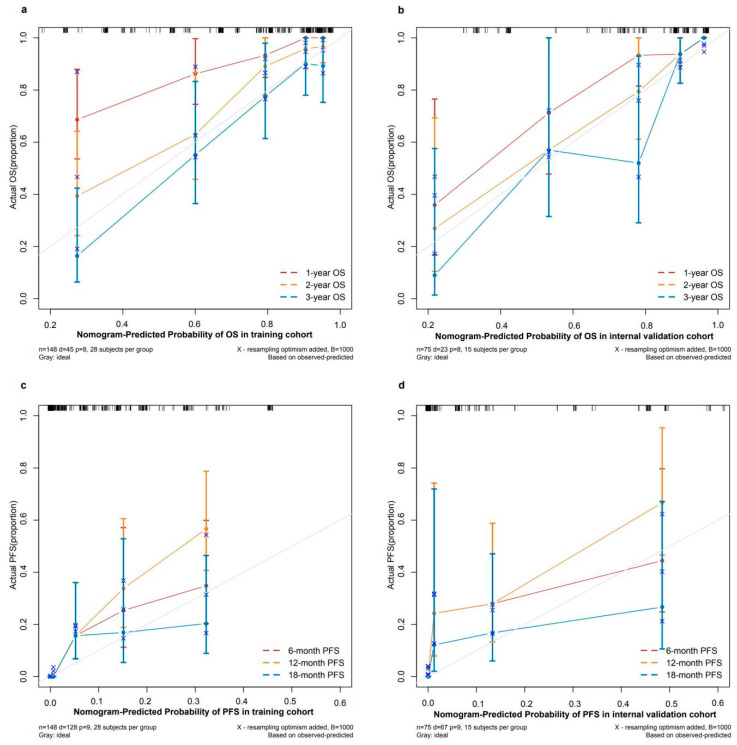
Calibration curves of the nomograms. (**a**) Calibration curves of nomogram for predicting 1-, 2-, 3-year OS in training cohort. (**b**) Calibration curves of nomogram for predicting 6-, 12-, 18-month PFS in internal validation cohort. (**c**) Calibration curves of nomogram for predicting 1-, 2-, 3-year OS in training cohort. (**d**) Calibration curves of nomogram for predicting 6-, 12-, 18-month PFS in internal validation cohort.

**Figure 6 pharmaceuticals-17-00373-f006:**
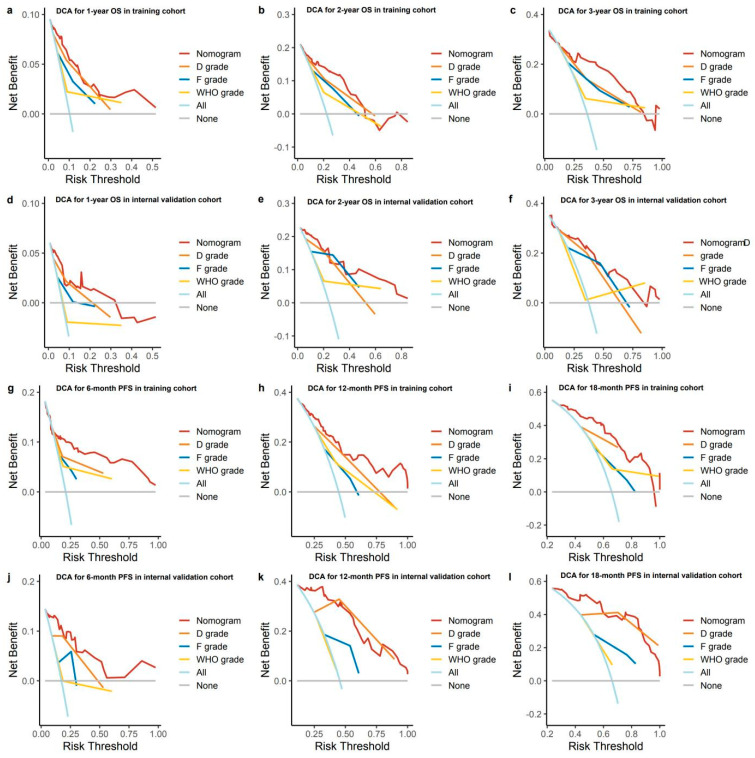
DCA of the nomograms, dual ^18^F−FDG and ^68^Ga−DOTANOC PETCT visual grade (D grade), semiquantitative ^18^F-FDG PETCT grade (F grade) and WHO pathological grading system (G grade) (**a**–**c**) DCA for predicting 1-, 2-, 3-year OS in training cohort. (**d**–**f**) Calibration curves of nomogram for predicting 1-, 2-, 3-year OS in internal validation cohort. (**g**–**i**) DCA for predicting 6-, 12-, 18-month PFS in training cohort. (**j**–**l**) DCA for predicting 6-, 12-, 18-month PFS in internal validation cohort.

**Table 1 pharmaceuticals-17-00373-t001:** Baseline characteristics of Training dataset and internal validation dataset.

Characteristics	Total	Training Dataset	Internal Validation Dataset	*p*-Value
	(*N* = 223)	(*N* = 148)	(*N* = 75)	
Age				
<55	113 (50.7%)	78 (52.7%)	35 (46.7%)	0.478
≥55	110 (49.3%)	70 (47.3%)	40 (53.3%)	
Sex				
Male	130 (58.3%)	80 (54.1%)	50 (66.7%)	0.097
Female	93 (41.7%)	68 (45.9%)	25 (33.3%)	
Primary tumor site				
Stomach	33 (14.8%)	25 (16.9%)	8 (10.7%)	0.439
Small intestine	51 (22.9%)	34 (23.0%)	17 (22.7%)	
Colorectum	139 (62.3%)	89 (60.1%)	50 (66.7%)	
Extrehepatic metastases				
No	32 (14.3%)	25 (16.9%)	7 (9.3%)	0.187
Yes	191 (85.7%)	123 (83.1%)	68 (90.7%)	
Therapy				
Without surgery	105 (47.1%)	72 (48.6%)	33 (44.0%)	0.606
After surgery	118 (52.9%)	76 (51.4%)	42 (56.0%)	
WHO Grade				
G1	42 (18.8%)	26 (17.6%)	16 (21.3%)	0.767
G2	161 (72.2%)	108 (73.0%)	53 (70.7%)	
G3	20 (9.0%)	14 (9.5%)	6 (8.0%)	
D				
D1	92 (41.3%)	58 (39.2%)	34 (45.3%)	0.514
D2	86 (38.6%)	61 (41.2%)	25 (33.3%)	
D3	45 (20.2%)	29 (19.6%)	16 (21.3%)	
F				
F1	115 (51.6%)	73 (49.3%)	42 (56.0%)	0.595
F2	62 (27.8%)	44 (29.7%)	18 (24.0%)	
F3	46 (20.6%)	31 (20.9%)	15 (20.0%)	
S				
S1	21 (9.4%)	14 (9.5%)	7 (9.3%)	0.986
S2	55 (24.7%)	36 (24.3%)	19 (25.3%)	
S3	147 (65.9%)	98 (66.2%)	49 (65.3%)	
FDG SUVmax				
Mean (SD)	4.32 (4.09)	4.31 (3.94)	4.34 (4.39)	0.967
Median [Min, Max]	3.40 [0, 23.8]	3.55 [0, 23.8]	3.20 [0, 20.2]	
SSA SUVmax				
Mean (SD)	14.0 (9.42)	13.4 (8.57)	15.1 (10.9)	0.243
Median [Min, Max]	13.5 [0, 49.3]	12.5 [0, 46.8]	15.0 [0, 49.3]	
NETPET score				0.023
P1	65 (29.1%)	37 (25.0%)	28 (37.3%)	
P2a	27 (12.1%)	21 (14.2%)	6 (8.0%)	
P2b	86 (38.6%)	61 (41.2%)	25 (33.3%)	
P3a	3 (1.3%)	3 (2.0%)	0 (0%)	
P3b	16 (7.2%)	11 (7.4%)	5 (6.7%)	
P4a	7 (3.1%)	1 (0.7%)	6 (8.0%)	
P4b	11 (4.9%)	9 (6.1%)	2 (2.7%)	
P5	8 (3.6%)	5 (3.4%)	3 (4.0%)	

D: dual ^18^F-FDG and ^68^Ga-DOTANOC PET-CT semiquantitive grade; F: single ^18^F-FDG PET-CT semiquantitive grade; S: single ^68^Ga-DOTANOC PET-CT semiquantitive grade.

**Table 2 pharmaceuticals-17-00373-t002:** Efficiency of different grading systems for predicting OS and PFS.

System	Overall Survival				Progression-Free Survival			
	C-Index (95% CI)	AIC	LR Test	R2	C-Index (95% CI)	AIC	LR Test	R2
D grade	0.763 (0.714–0.812)	603.02	56.82	0.237	0.724 (0.690–0.758)	1672.51	96.84	0.345
F grade	0.727 (0.770–0.785)	621.37	38.46	0.167	0.630 (0.593–0.667)	1735.05	34.3	0.143
S grade	0.566 (0.500–0.634)	655.19	4.65	0.022	0.556 (0.514–0.598)	1761.48	7.87	0.035
WHO grade	0.650 (0.592–0.709)	629.26	30.58	0.135	0.592 (0.552–0.631)	1769.43	−0.08	0

D: dual ^18^F-FDG and ^68^Ga-DOTANOC PET-CT semiquantitive grade; F: single ^18^F-FDG PET-CT semiquantitive grade; S: single ^68^Ga-DOTANOC PET-CT semiquantitive grade.

**Table 3 pharmaceuticals-17-00373-t003:** Uni- and Multivariate Cox Analyses for OS in Training Cohort and Internal Validation Cohort.

Characteristics	Training Cohort				Internal Validation Cohort			
	Univariate Cox Regression		Multivariate Cox Regression		Univariate Cox Regression		Multivariate Cox Regression	
	HR (95% CI)	*p*-Value	HR (95% CI)	*p*-Value	HR (95% CI)	*p*-Value	HR (95% CI)	*p*-Value
Age								
<55	Reference		Reference		Reference		Reference	
≥55	2.21 (1.21–4.05)	0.010	1.56 (0.8–3.03)	0.188	5.70 (1.93–16.79)	0.002	3.79 (1.19–12.06)	0.024
Extrehepatic metastases								
No	Reference		Reference		Reference		Reference	
Yes	2.27 (0.89–5.78)	0.087	1.3 (0.49–3.45)	0.592	0.80 (0.19–3.43)	0.764	0.51 (0.09–2.71)	0.427
WHO Grade								
G1	Reference		Reference		Reference		Reference	
G2	3.17 (0.97–10.39)	0.057	2.22 (0.64–7.74)	0.209	1.20 (0.39–3.68)	0.744	0.91 (0.27–3.04)	0.881
G3	14.41 (3.97–52.3)	<0.001	4.21 (1.03–17.25)	0.045	9.38 (2.35–37.46)	0.002	3.25 (0.69–15.46)	0.138
D								
D1	Reference		Reference		Reference		Reference	
D2	3.82 (1.52–9.63)	0.005	3.66 (1.41–9.5)	0.008	6.05 (1.65 −22.09)	0.007	4.7 (1.27–18.49)	0.020
D3	15 (5.78–38.91)	<0.001	6.12 (2.02–18.57)	<0.001	11.79 (3.23–43.11)	<0.001	6.34 (1.59–25.30)	0.009
F								
F1	Reference		Reference		Reference		Reference	
F2	2.92 (1.41–6.06)	0.004	1.87 (0.85–4.13)	0.121	5.6 (1.86–16.83)	0.002	2.83 (0.72–11.11)	0.137
F3	5.39 (2.74–12.86)	<0.001	2.55 (1.02–6.39)	0.045	7.87 (2.61–23.7)	<0.001	3.05 (0.75–12.51)	0.121

D: dual ^18^F-FDG and ^68^Ga-DOTANOC PET-CT semiquantitive grade; F: single ^18^F-FDG PET-CT semiquantitive grade.

**Table 4 pharmaceuticals-17-00373-t004:** Uni- and Multivariate Cox Analyses for PFS in Training Cohort and Internal Validation Cohort.

Characteristics	Training Cohort				Internal Validation Cohort			
	Univariate Cox Regression		Multivariate Cox Regression		Univariate Cox Regression		Multivariate Cox Regression	
	HR (95% CI)	*p*-Value	HR (95% CI)	*p*-Value	HR (95% CI)	*p*-Value	HR (95% CI)	*p*-Value
Age								
<55	Reference		Reference		Reference			
≥55	1.72 (1.21–2.44)	0.003	1.54 (1.05–2.25)	0.027	3.30 (1.93–5.63)	<0.001	3.57 (1.98–6.44)	<0.001
Extrehepatic metastases								
No	Reference		Reference		Reference			
Yes	1.50 (0.93–2.43)	0.097	1.2 (0.72–1.99)	0.477	1.14 (0.49–2.64)	0.768	1.45 (0.54–3.9)	0.467
Therapy								
Without surgery	Reference		Reference		Reference			
After surgery	0.64 (0.45–0.91)	0.013	0.76 (0.52–1.11)	0.159	0.69 (0.42–1.12)	0.137	0.66 (0.38–1.15)	0.142
WHO Grade								
G1	Reference		Reference		Reference			
G2	1.82 (1.11–2.99)	0.018	1.62 (0.95–2.76)	0.076	1.46 (0.78–2.73)	0.231	1.11 (0.57–2.16)	0.753
G3	8.14 (3.93–16.84)	<0.001	5.2 (2.35–11.54)	<0.001	3.64 (1.35–9.83)	0.011	2.41 (0.75–7.8)	0.142
D								
D1	Reference		Reference		Reference			
D2	2.11 (1.40–3.17)	<0.001	2.13 (1.38–3.31)	<0.001	3.36 (1.88–5.99)	<0.001	2.99 (1.42–6.29)	0.004
D3	7.93 (4.72–13.31)	<0.001	7.33 (3.59–15)	<0.001	15.24 (7.21–32.23)	<0.001	25.1 (8.57–73.46)	<0.001
F								
F1	Reference		Reference		Reference			
F2	1.93 (1.28–2.90)	0.002	1.42 (0.92–2.21)	0.114	2.52 (1.41–4.51)	0.002	1.7 (0.8–3.6)	0.164
F3	2.34 (1.48–3.68)	<0.001	0.74 (0.4–1.38)	0.347	3.23 (1.64–6.36)	<0.001	0.63 (0.25–1.61)	0.337

D: dual ^18^F-FDG and ^68^Ga-DOTANOC PET-CT semiquantitive grade; F: single ^18^F-FDG PET-CT semiquantitive grade.

**Table 5 pharmaceuticals-17-00373-t005:** Efficiency of Different Grading System for Predicting OS and PFS in the Training Cohort.

System	Overall Survival				Progression-Free Survival			
	C-Index (95% CI)	AIC	LR Test	R2	C-Index (95% CI)	AIC	LR Test	R2
Nomogram	0.810 (0.767–0.874)	354.44	54.24	0.32	0.741 (0.692–0.789)	993.3	83.78	0.433
D grade	0.759 (0.700–0.821)	355.16	39.52	0.252	0.700 (0.653–0.748)	1009.36	53.71	0.305
F grade	0.710 (0.637–0.783)	373.17	21.35	0.146	0.617 (0.569–0.666)	1046.31	16.76	0.107
WHO grade	0.661 (0.590–0.732)	373.02	21.66	0.147	0.613 (0.569–0.657)	1036.43	26.74	0.165

D: dual ^18^F-FDG and ^68^Ga-DOTANOC PET-CT semiquantitive grade; F: single ^18^F-FDG PET-CT semiquantitive grade.

**Table 6 pharmaceuticals-17-00373-t006:** Efficiency of Different Grading System for Predicting OS and PFS in the Internal Validation Cohort.

System	Overall Survival				Progression-Free Survival			
	C-Index (95% CI)	AIC	LR Test	R2	C-Index (95% CI)	AIC	LR Test	R2
Nomogram	0.849 (0.781–0.849)	158.03	35.89	0.419	0.824 (0.778–0.871)	410.79	76.17	0.639
D grade	0.779 (0.698–0.860)	162.29	19.23	0.249	0.772 (0.729–0.815)	425.51	47.45	0.47
F grade	0.760 (0.664–0.855)	164.11	17.81	0.233	0.649 (0.594–0.704)	457.92	15.4	0.182
WHO grade	0.633 (0.527–0.738)	171.19	10.73	0.147	0.555 (0.478–0.633)	467.38	5.58	0.072

D: dual ^18^F-FDG and ^68^Ga-DOTANOC PET-CT semiquantitive grade; F: single ^18^F-FDG PET-CT semiquantitive grade.

## Data Availability

The datasets analyzed during the current study are available in the Clinical Pathology Database and Nuclear Medicine Database of the First Affiliated Hospital of Sun Yat-sen University.

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
