# Peer review of "Evaluating Prognosis of Gastrointestinal Metastatic Neuroendocrine Tumors: Constructing a Novel Prognostic Nomogram Based on NETPET Score and Metabolic Parameters from PET/CT Imaging"

_pharmaceuticals, 2024, doi:10.3390/ph17030373_

Round 1

Reviewer 1 Report

Comments and Suggestions for Authors

This study effectively compares the prognostic performance of NETPET scores derived from 68Ga-DOTANOC and 18F-FDG PET-CT in metastatic GI-NET patients. Nomograms developed from these scores accurately predict progression-free survival (PFS) and overall survival (OS), demonstrating superior prognostic value over traditional PET-CT metabolic parameters.

Recommendation: The introduction is too short; please add more details and literature to support your study and provide a comparative introduction of emission tomography-computed tomography (PET-CT) with already existing techniques used in gastrointestinal metastatic neuroendocrine tumor detection.

Line 44. Please give more detail about neuroendocrine neoplasms (NENs) in the introduction.

Include an introduction to Gallium-68 DOTANOC (68Ga-DOTANOC) and Fluorine-18 fluorodeoxyglucose (18F-FDG) Positron Emission Tomography-Computed Tomography (PET-CT) in the introduction section, explaining their mechanisms of action briefly.

Please add the conclusion section to the manuscript.

Reviewer 2 Report

Comments and Suggestions for Authors

The manuscript presents a significant and innovative study with the potential to advance the prognostic evaluation of gastrointestinal neuroendocrine tumors (GI-NETs) using PET/CT imaging parameters. The construction and validation of a novel prognostic nomogram represent a valuable contribution to the field. However, to ensure the manuscript meets the high standards of Pharmaceutics, the following revisions are recommended:

Major Concerns:

1.      Methodological Clarification: The manuscript needs to provide a more detailed rationale for the patient selection criteria and the choice of specific metabolic parameters from PET/CT imaging for prognostic modeling. This is crucial for understanding the study's applicability and the significance of its findings.

2.      Statistical Analysis Rigor: Additional information on measures taken to prevent overfitting and a sensitivity analysis to assess the model's robustness across different patient subgroups are required to ensure the reliability and generalizability of the study's conclusions.

3.      Clinical Utility Discussion: A more detailed discussion on the integration of the nomogram into clinical practice is needed. This should include potential impacts on treatment decisions and patient outcomes, supplemented by case studies or scenarios demonstrating the nomogram's application in real-world settings.

Minor Concerns:

1.      Supplementary Material Presentation: The supplementary figures and tables should be reviewed for quality and clarity. Additionally, the Kaplan-Meier survival analysis and its implications for patient care require a more in-depth discussion within the context of the study's findings.

2.      Terminology Consistency: Ensure all acronyms are defined at their first use in the manuscript, and maintain consistency in terminology and methodological descriptions across the main manuscript and supplementary materials.

3.      Address any typographical, grammatical, or formatting issues to meet the high editorial standards expected by the Pharmaceutics journal.

Recommendations for Revision:

·         Address each point of concern in detail, providing additional justification, clarification, and evidence as appropriate.

·         Ensure the revised manuscript clearly articulates the study's contribution to the existing body of knowledge, emphasizing its novelty and potential clinical impact.

·         Enhance the presentation and accessibility of both the manuscript and supplementary materials, ensuring they work cohesively to support the study's conclusions.

Reviewer 3 Report

Comments and Suggestions for Authors

The article "Evaluating Prognosis of Gastrointestinal Metastatic Neuroendocrine Tumors: Constructing a Novel Prognostic Nomogram Based on NETPET Score and Metabolic Parameters from PET/CT Imaging" presents a retrospective study on 223 patients with gastrointestinal neuroendocrine tumors (GI-NETs). It compares the prognostic performance of NETPET scores, derived from dual-scan PET-CT using 68Ga-DOTANOC and 18F-FDG, with PET-CT metabolic parameters. The study constructs and validates a nomogram for predicting progression-free survival (PFS) and overall survival (OS). Based on multivariable Cox models and LASSO regression, the nomograms showed superior prognostic performance with AUCs >0.8. The study concludes that the new classification based on the NETPET score provides the best prediction for patient OS/PFS and that PET-CT-based nomograms offer accurate forecasts for these outcomes.

Following are a few shortcomings of the article:

Methodology

1. Retrospective Design: The study's retrospective nature inherently limits the ability to control for confounding variables and biases that may have influenced the results. While retrospective studies are valuable for hypothesis generation and exploring associations, they are less robust than prospective studies in establishing causality.

2. Selection Criteria: The inclusion and exclusion criteria, while necessary for defining the study population, may also limit the generalizability of the findings. Specifically, excluding patients with primary tumors outside the gastrointestinal tract or with unknown primary locations could omit a subset of patients with neuroendocrine tumors (NETs) relevant to the broader NET patient population.

3. Imaging Protocols: The reliance on specific PET-CT imaging protocols (68Ga-DOTANOC and 18F-FDG) may not be universally available or standardized across different institutions. This could affect the reproducibility and applicability of the grading systems and nomograms developed in the study.

4. Standardization of Imaging Analysis: The study's reliance on specific PET/CT imaging protocols and the calculation of SUVmax values could be influenced by inter-operator variability and the lack of standardized imaging analysis protocols across different centers. This variability could affect the consistency and reproducibility of the NETPET scores and subsequent grading systems.

5. Use of X-tile Software for Stratification: The stratification of NETPET scores, 18F-FDG PET-CT imaging SUVmax, and 68Ga-DOTANOC PET-CT imaging SUVmax into three tiers using X-tile software is innovative. However, the reliance on this specific software for categorization could limit the accessibility and applicability of the grading systems, especially in settings where this software is not readily available.

 Results

1. Statistical Power and Sample Size: The study involved 223 patients, which, while substantial, may still limit the statistical power for detecting smaller effect sizes, especially in subgroup analyses. The division into training and validation cohorts further reduces the sample size for each analysis.

2. Complexity of Grading Systems: The introduction of three grading systems based on NETPET scores and SUVmax could be a strength. However, the complexity and requirement for specific software (e.g., X-tile) for categorization may limit their practical utility in clinical settings.

3. Predictive Performance: While the nomograms showed superior prognostic performance with AUCs >0.8, the comparison with existing prognostic tools, such as the WHO histological grading system, was not detailed. A more thorough comparison, including limitations and advantages over existing methods, would provide a more precise context for the nomograms' utility.

4. Lack of External Validation: The study divides the cohort into a training set and an internal validation set, a standard approach for developing predictive models. However, the absence of an external validation cohort to test the nomograms limits the ability to assess their generalizability and performance in a broader, independent patient population.

5. Detailed Analysis of Subgroups: While the study provides an overall assessment of the grading systems' predictive performance, a more detailed analysis of their performance across different subgroups (e.g., based on primary tumor location, histological grade, or treatment modality) could offer insights into their utility in specific clinical scenarios.

 Discussion

1. Generalizability: The study's findings are based on a specific patient population from a single institution, which may limit their generalizability to other populations or settings. Differences in patient demographics, healthcare systems, and treatment approaches could influence the applicability of the results.

2. Future Research Directions: While the study concludes with developing promising prognostic tools, it could have provided more specific recommendations for future research, such as prospective validation studies, comparisons with other predictive models, or exploration of the nomograms' impact on clinical decision-making.

3. Clinical Utility: The discussion could benefit from a deeper exploration of how these nomograms can be integrated into clinical practice, including potential challenges and the need for clinician education on interpreting and applying the new grading systems.

4. Integration with Existing Prognostic Models: The discussion could have elaborated on how the newly developed nomograms and grading systems could be integrated with existing prognostic models and clinical guidelines for managing patients with gastrointestinal metastatic neuroendocrine tumors. This would help clinicians understand how to use these tools with other prognostic indicators.

5. Consideration of Treatment Effects: The study's discussion section could have addressed the potential impact of different treatment modalities on the prognostic performance of the nomograms. Given that treatment plans were decided by the hospital's NETMDT team and could include surgical or medical treatments, understanding how these interventions might influence prognosis and the nomograms' predictive accuracy would be valuable.

6. Addressing Limitations of PET/CT Imaging: While PET/CT imaging provides valuable metabolic and anatomical information, its limitations, such as radiation exposure, cost, and availability, should have been discussed. Acknowledging these limitations and exploring alternative or complementary imaging modalities could provide a more comprehensive view of the challenges in the prognostic evaluation of these tumors.

In conclusion, the article significantly contributes to the field by developing novel prognostic tools for gastrointestinal metastatic neuroendocrine tumors. However, addressing the additional shortcomings could enhance the findings' robustness, applicability, and clinical utility.

Comments on the Quality of English Language

minor

Reviewer 4 Report

Comments and Suggestions for Authors

General Comments:

The manuscript by Liu et al. presents a comprehensive study aimed at developing a novel prognostic nomogram for patients with gastrointestinal metastatic neuroendocrine tumors (GI-NETs) using PET/CT imaging parameters combined with NETPET scores. This research addresses a significant gap in the current prognostic evaluation methods for GI-NETs, focusing on the integration of metabolic imaging data and clinical-pathological features to refine prognosis prediction. The methodological approach, including a well-defined patient cohort, rigorous imaging analysis, and statistical validation, underpins the study's significance. However, certain aspects require further clarification and improvement to enhance the manuscript's contribution to the field.

Specific Comments:

  1. Methodology:
    • The study's methodology is robust, with clear inclusion and exclusion criteria for patient selection, detailed imaging analysis protocols, and comprehensive statistical evaluations. However, the manuscript would benefit from a more detailed explanation of the NETPET scoring system's rationale and its integration into the prognostic model.
    • One significant aspect that could further enhance the manuscript's contribution to the NET prognostic evaluation field is the inclusion of a comparative analysis between the NETPET score and circulating biomarkers, particularly Chromogranin A (CgA) or NSE, in terms of their prognostic capabilities. This comparison could offer a more comprehensive understanding of the NETPET score's relative performance and potential advantages over traditional biomarkers in predicting patient outcomes.

  1. Results:

·       The results are clearly presented, with the novel grading systems (D, F, S grades) demonstrating significant prognostic value. The superiority of the D grading system in discriminating OS and PFS is a critical finding. The authors should consider discussing any potential limitations of the D grading system and its applicability in clinical practice.

·        Comparative Analysis with Chromogranin A: The authors are encouraged to incorporate a section that compares the prognostic capabilities of the NETPET score with circulating markers like Chromogranin A (CgA) (see PMID: 30325865) or multigene liquid biopsies (PMID: 31841822, PMID: 30098712). If they didn’t collect these data they even make this comparison based on a literature review or, if data are available, a retrospective analysis within the study cohort.

  1. Discussion and Interpretation:
    • The discussion effectively highlights the study's significance and the potential clinical implications of the findings. It would be beneficial to include a more comprehensive comparison with existing prognostic models and discuss the implications of adopting this nomogram in routine clinical practice.
    • Comparative Analysis with Chromogranin A: It would be informative to discuss the sensitivity, specificity, and predictive value of these markers in comparison with generally used circulating markers and eventually addressing the Limitations of CgA (PMID: 31266019)
    • Limitations: The authors acknowledge the study's limitations, including its retrospective design and single-center nature. Future studies should aim for prospective validation and involve multi-center collaborations to enhance the nomogram's generalizability and clinical utility.

Specific Suggestions:

 - An analysis comparing NETPET scores with circulating biomarkers should also delve into the clinical implications of using one over the other or in combination. The authors could explore scenarios where the NETPET score might offer clearer prognostic insights or complement CgA findings, particularly in managing patients with complex or concomitant diseases or therapies, conditions in which CgA may fail (see PMID: 18762462).

- Finally, the manuscript could benefit from recommendations for future research directions, especially prospective studies designed to directly compare the prognostic accuracy of PET/CT-based scores with circulating biomarkers. Highlighting the potential for combining these markers to enhance prognostic models could also be an area of interest.

Comments on the Quality of English Language

The manuscript is generally well-written and articulate

Round 2

Reviewer 4 Report

Comments and Suggestions for Authors

The paper makes a valuable contribution to the field of oncology, particularly in the management of GI-NETs, by providing a novel and clinically relevant prognostic tool. Further research is warranted to establish its applicability and effectiveness in broader clinical settings. No further requests

Comments on the Quality of English Language

The quality of English in the paper is generally good